

# Mapping avalanches with satellites – evaluation of performance and completeness

Elisabeth D. Hafner[1], Frank Techel[1,3], Silvan Leinss[2], Yves Bühler[1]

[1]WSL Institute for Snow and Avalanche Research SLF, Davos Dorf, 7260, Switzerland
[2]Institute of Environmental Engineering, ETH Zurich, Zurich, Switzerland
[3] Department of Geography, University of Zurich, Zurich, Switzerland

*Correspondence to*: Elisabeth Hafner (elisabeth.hafner@slf.ch)

**Abstract.**

The spatial distribution and size of avalanches are essential parameters for avalanche warning,
avalanche documentation, mitigation measure design and hazard zonation. Despite its importance, this
information is incomplete today and only available for limited areas and limited time periods. Manual
avalanche mapping from satellite imagery has recently been applied to reduce this gap achieving
promising results. However, their reliability and completeness were not yet verified satisfactorily.
In our study we attempt a full validation of the completeness of visually detected and mapped
avalanches from optical SPOT-6, Sentinel-2 and radar Sentinel-1 imagery. We examine manually
mapped avalanches from two avalanche periods in 2018 and 2019 for an area of approximately 180 km$^2$
around Davos, Switzerland relying on ground- and helicopter-based photographs as ground truth. For
the quality assessment, we investigate the Probability of Detection (POD) and the Positive Predictive
Value (PPV). Additionally, we relate our results to conditions which potentially influence avalanche
detection in the satellite imagery.
We statistically confirm the high potential of SPOT for comprehensive avalanche mapping for selected
periods (POD= 0.74, PPV = 0.88) as well as the reliability of Sentinel-1 for the mapping of larger
avalanches (POD= 0.27, PPV = 0.87). Furthermore, we proof that Sentinel-2 is unsuitable for the
mapping of most avalanches due to its spatial resolution (POD= 0.06, PPV = 0.81). Because we could
apply the same reference avalanche events for all three satellite mappings, our validation results are
robust and comparable. We demonstrate that satellite-based avalanche mapping has the potential to fill
the existing avalanche documentation gap over large areas, making alpine regions safer.

## 1   Introduction

Where and when avalanches occur, and what size and destructive potential they have, is key
information to mitigate avalanche hazard in snow-covered mountain regions. Several applications
depend on such information:

- Avalanche warning: validation of the avalanche forecast (Bühler et al., 2019; Meister, 1994)



- Hazard zoning: complementation of existing cadasters and validation of the hazard zones (Bühler et al., 2018; Rudolf-Miklau et al., 2014)
- Hazard mitigation measures: validation of effectiveness and planning of new infrastructure (Rudolf-Miklau et al., 2014; Margreth and Romang, 2010)
- Forestry: identification of potential forest damage and examination of protective functions (Bebi et al., 2009; Feistl et al., 2015)
- Risk management: categorization and understanding of the severity of events and estimation of cost-effective solutions (Fuchs et al., 2005; Bründl and Margreth, 2015)
- Numerical simulations: validation of avalanche models (Christen et al., 2010; Sampl and
Zwinger, 2004; Bühler et al., 2011)

Nevertheless, information on avalanche occurrence is only available for limited areas and timespans. This means that most avalanche events are not reported and therefore not captured in any database or cadaster, and particularly within poorly accessible regions (Bühler et al., 2019).

Remote sensing technology is increasingly used to record and map avalanche occurrences with consistent methodology and continuous spatial coverage over large regions. Optical data from airplanes and satellites with high to very high spatial resolution (0.1 – 1.5 m) have been successfully used in the past to manually or semi-automatically map avalanches (Bühler et al., 2009; Lato et al., 2012;
Eckerstorfer et al., 2016; Korzeniowska et al., 2017; Bühler et al., 2019). High to very high spatial resolution optical data have mostly limited coverage and a low temporal resolution as they are usually available upon request only. Furthermore, they are often costly and depend on cloud free conditions. Optical satellites under free and open data policy with a high temporal resolution but lower spatial resolution like Sentinel-2 have only been tested briefly for snow avalanche detection or were used to
complement Sentinel-1 investigations (Nolting et al., 2018; Abermann et al., 2019). For the documentation of individual avalanche events, unmanned aerial systems (UASs) equipped with optical cameras can flexibly provide detailed information but they are not able to cover larger regions (Bühler et al., 2017; Eckerstorfer et al., 2016).

In the microwave spectrum, radar sensors operate independently of light and weather conditions. Radar sensors can detect the increased roughness of the snow surface caused by avalanches (Eckerstorfer and Malnes, 2015, Leinss et al. 2020). Radar satellites, like RadarSat, TerraSAR-X, and Sentinel-1, have been successfully applied for avalanche mapping in various regions (Eckerstorfer and Malnes, 2015; Vickers et al., 2016; Eckerstorfer et al., 2017; Wesselink et al., 2017; Abermann et al., 2019; Leinss et
al., 2020). Selective verification has shown that radar underestimates the avalanche activity to an unknown extent (Eckerstorfer et al., 2017). Often only parts of the avalanches are mapped, and Sentinel-1 misses most small avalanches due to the limited spatial resolution (Leinss et al., 2020).

For the mapping of avalanches, or parts thereof, change detection and unsupervised object classification
(Vickers et al., 2016), semi-automated object- based approaches (Korzeniowska et al., 2017; Lato et al., 2012), automated change detection approaches (Wesselink et al., 2017; Nolting et al., 2018; Eckerstorfer et al., 2019) as well as manual mapping (Bühler et al., 2019; Abermann et al., 2019;



Eckerstorfer et al., 2015) and combinations of manual and automatic mapping (Leinss et al., 2020) have been used.

As consistent avalanche detection using satellite data is becoming increasingly important, the identification of its performance and reliability is essential. To do so, we assess the completeness of visually detected and manually mapped avalanches, using three different sensors, which have recently been used to detect avalanches (e.g. Eckerstorfer and Malnes, 2015; Leinss et al., 2020; Bühler et al., 2019; Nolting et al., 2018; Abermann et al., 2019):

- Optical SPOT-6, commercial, 1.5 m spatial resolution
- Radar Sentinel-1, open access, 10 m spatial resolution
- Optical Sentinel-2, open access, 10 m spatial resolution

As validation data, we rely on photographs taken from the ground and from helicopters to document two extreme avalanche situations in 2018 and 2019, in Davos in Eastern Switzerland. We compare the completeness of avalanches detected with the three sensors by answering the following two research questions:

1. Of the avalanches identified in the ground-truth, how many were correctly detected by a human in the satellite data?
2. If a human visually detected an avalanche in the satellite data, how often was there an avalanche?

Furthermore, we investigate these findings in relation to conditions which potentially influence avalanche detection in satellite imagery. To do so we consider the size of the mapped avalanche from all approaches, the illumination conditions in optical SPOT-6 (SPOT hereafter) data and the predominantly detected parts of the avalanches in radar data. Finally, we highlight the potential and the limitations of a well-established, multi-year data set of mapped avalanches as an existing data source for validation.

## 2 Area and Datasets

### 2.1 Study area and validation period

Our study area of approximately 180 km$^2$ is located around Davos, Switzerland (Figure 1). 25 % of the area are considered avalanche release areas according to the release area definitions introduced by Bühler et al. (2018). The study area comprises the main valley and parts of three inhabited side valleys (Flüela, Dischma, Sertig), as well as the surrounding mountains and covers an elevation range from 1450 m to 2981 m a.s.l. In January 2018, 93 % of the study area were covered by SPOT satellite imagery ordered for rapid mapping (Bühler et al., 2019). The 7 % which were missed were excluded from our study and are shaded in red in Figure 1. In January 2019 the entire area was covered by SPOT.



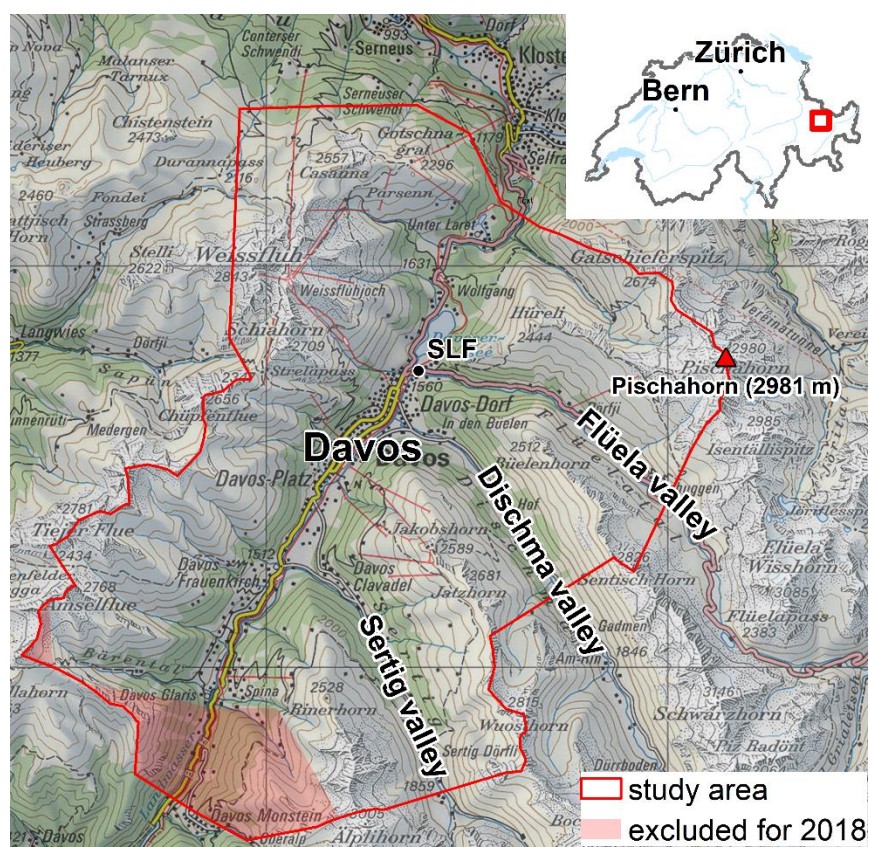

**Figure 1: The location of the validation area with the side valleys the SLF and the highest point. The red-shaded area was omitted in 2018 due to lack of SPOT satellite data (Pixmap © 2019 swisstopo (5 704 000 000), reproduced by permission of swisstopo (JA100118)).**

For the validation, we considered two periods with high avalanche activity (Figure 2):

- 20 to 24 January 2018, referred to as 2018: Following several snowfalls in January, about 150 cm of snow fell in 65 hours. This was followed by rain up to 2000 m a.s.l. Consequently, numerous wet-snow, dry-snow and gliding avalanches released.
- 13 to 16 January 2019, referred to as 2019: Following several snowfalls in the two weeks before, heavy snowfall brought about 100 cm of new snow in 60 hours. This resulted in mostly dry-snow avalanches, some with a destructive powder blast.

In both situations, danger levels 4 (high) and 5 (very high) on the five-level ordinal European Avalanche Danger Scale (WSL, 2019) were forecasted in the study area and avalanches of all sizes released.





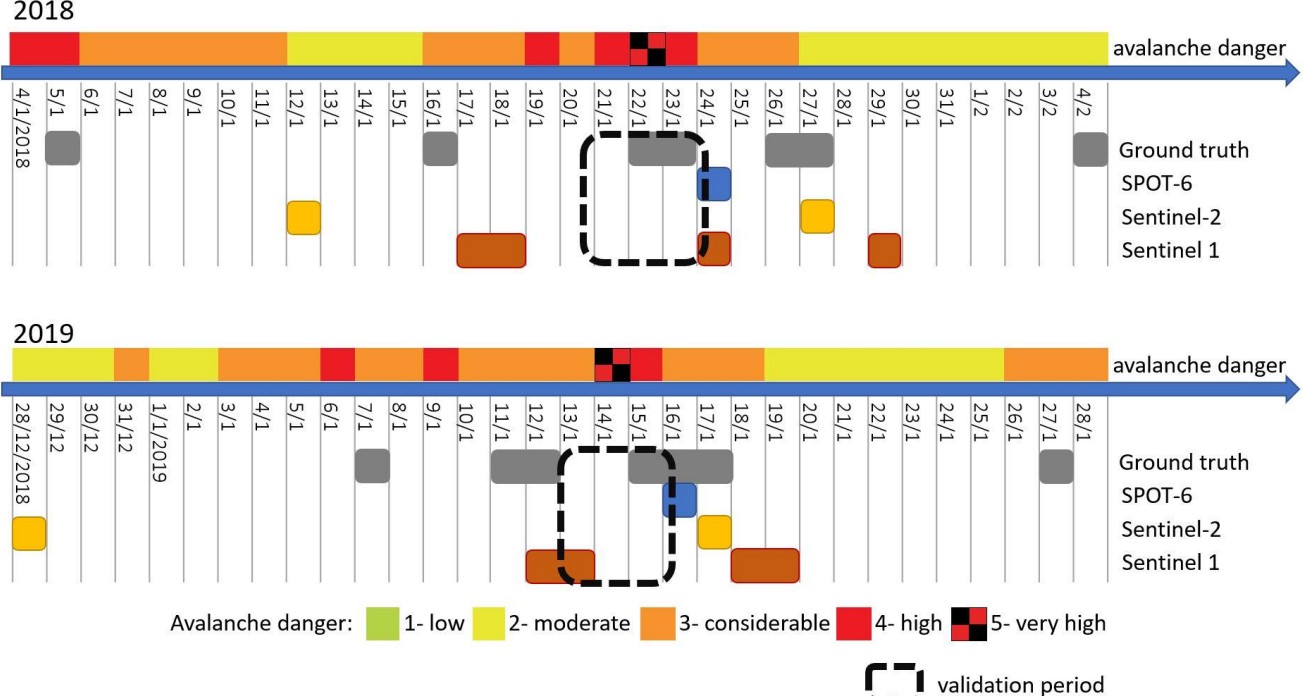

**Figure 2: Temporal overview of the two validation periods (marked by the dashed, black squares) and the respective days, when ground truth images and satellite data were captured (marked by colored boxes). On top the highest avalanche danger level forecasted in the validation area is shown for each day.**

5 ## 2.2 Satellite data

High spatial resolution (1.5 m) SPOT satellite imagery were acquired after the two validation periods on request (Figure 2, Table 1). From operationally acquired medium-spatial resolution (10 m) Sentinel-1 (Table 2) and Sentinel-2 (Table 1) acquisitions we selected images from before and after the validation period.

**Table 1: Properties of the data acquired by the optical sensors (SPOT 6 and Sentinel-2).**

|  | Satellite | Acquisition time (UTC) | Inclination angle | Spatial resolution of used bands (m) | Spectral resolution of used bands |
|---|---|---|---|---|---|
| 2018 | SPOT-6 | 2018-01-24 10:03 | 11.3° and 21.5° | 1.5 m panchromatic 6 m multispectral | Blue: 455 nm–525 nm Green: 530 nm–590 nm Red: 625 nm–695 nm Near Infrared: 760 nm–890 nm |
| 2019 | SPOT-6 | 2019-01-16 10:04 | 16,7° and 23° | | |
| 2018 | S2A | 2018-01-12 10:22 | ~0° | 10 m | Central wave-length 3 (Green): 559.8 nm 4 (Red): 664.6 nm 8 (Infrared): 832.8 nm |
| | S2B | 2018-01-27 10:24 | | | |
| 2019 | S2A | 2018-12-28 10:24 | | | |
| | S2A | 2019-01-17 10:23 | | | |





**Table 2: Properties of the Sentinel-1 radar acquisitions.**

| Satellite | Acquisition time (UTC) | orbit | Mode | Pixel spacing (m) | Inc. angle θ | Polarizations |
|---|---|---|---|---|---|---|
| S1A | 2018-01-17 05:26 | 168 des | IW | 10 x 10 | 42.8° | VV, VH |
| S1A | 2018-01-18 17:15 | 15 asc | IW | 10 x 10 | 41.9° | VV, VH |
| S1A | 2018-01-29 05:26 | 168 des | IW | 10 x 10 | 42.8° | VV, VH |
| S1B | 2018-01-24 17:14 | 15 asc | IW | 10 x 10 | 41.9° | VV, VH |
| S1A | 2019-01-12 05:26 | 168 des | IW | 10 x 10 | 42.8° | VV, VH |
| S1A | 2019-01-13 17:15 | 15 asc | IW | 10 x 10 | 41.9° | VV, VH |
| S1B | 2019-01-18 05:26 | 168 des | IW | 10 x 10 | 42.8° | VV, VH |
| S1B | 2019-01-19 17:14 | 15 asc | IW | 10 x 10 | 41.9° | VV, VH |

## Data preprocessing- Optical data

We refrained from atmospheric corrections because they are not necessary for avalanche detection as
atmospheric effects are relatively minor for most regions in winter since the water content of the
atmosphere is typically low (Nolin, 2010). SPOT data was cloud free for both years. For Sentinel-2 in
2019, about 7% of the validation area on the post-event image was hidden by clouds. Because of the
reliance on manual mapping we refrained from cloud pre-processing.

*SPOT*
SPOT imagery was delivered with type "Primary", and pan-sharpened in full radiometric resolution (12
bit). The data was oriented using bundle block adjustment and orthorectified by swisstopo based on the
high-quality terrain model swissALTI3D resampled to 5 m (swisstopo, 2018). In addition to automated
tie-point generation, ground control points (GCPs) were digitized manually. The achieved accuracy
(RMSE) of the GCPs achieved a localization accuracy of better than 2 m in X and Y (Bühler et al.,
2019).

*Sentinel-2*
We composed the bands 3, 4 and 8 of the Sentinel-2 level 1C products into one false-color image with
10 m resolution. Orthorectification for Sentinel-2 level 1C relies on the 90-m resolution model Planet-
DEM-90 (https://sentinel.esa.int/web/sentinel/user-guides/sentinel-2-msi/definitions). Ressl and Pfeifer
(2014) found an approximate accuracy of location of ±10 m (i.e. one Sentinel-2 pixel). No additional
corrections of orthorectification were applied.

## Data preprocessing- Radar data

*Sentinel-1*
For processing, we followed the steps described in Leinss et al. (2020) but added local resolution
weighting (LRW; Small, 2012) to optimize the spatial resolution and to minimize terrain shadow and
layover effects. For LRW, two acquisitions from orbits with opposite view directions (ascending
looking east and descending looking west) were combined using a weighted average based on the local,
terrain dependent resolution of every pixel. Table 2 lists the set of pre- and post-event images used for
the two avalanche periods in 2018 and 2019.





The coherent imaging method of the synthetic aperture radar (SAR) system requires some spatial averaging to reduce radar speckle and to improve the radiometric accuracy of the backscatter intensity. The native resolution of the single-look-complex (SLC) interferometric wide swath mode (IW) images of Sentinel-1 is about 3 x 23 m (slant range x azimuth), provided at a slant-range pixel spacing of 2.3 x

14.1 m (Bourbigot et al., 2016). To avoid loss of resolution we averaged (multi-looked) the images with a relatively small window of 2 x 1 pixels (rg x az). Then we averaged the backscatter intensity ($\beta_0$) of both polarizations, VV and VH, scaled in dB to reduce the multiplicative speckle noise.

As LRW requires extremely precise geocoding on the sub-pixel level we co-registered the measured
backscatter intensity with the backscatter intensity simulated using a digital elevation model. We then orthorectified the measured and simulated backscatter images, sampled at a slant range resolution of 4.6 x 14.1 m (corresponds to a resolution of 6.9 x 14.1 m when projected on horizontal terrain), to a 10 x 10 m pixel spacing on the ground. For orthorectification (= geometric terrain correction), the Swiss Alti3D downsampled at 30 m resolution was used (swisstopo, 2018). Bilinear interpolation steps during
coregistration, orthorectification, and collocation slightly reduced the spatial resolution.

The orthorectified radar images were then radiometrically terrain corrected (Small, 2011) with the simulated intensity ($\beta_0^{TC} = \beta_0 / \beta_{sim}$) to remove the terrain-dependent illumination bias. LRW was applied on the backscatter signal of ascending (asc) and descending (des) acquisitions (in dB) using the
simulated intensity as weight ($w_{asc} = \beta_{sim,asc}$, $w_{des} = \beta_{sim,des}$):

$$\beta_0^{TC,LRW} = [\beta_{0,asc}^{TC} / w_{asc} + \beta_{0,des}^{TC} / w_{des}] / (w_{asc} + w_{des})$$

LRW optimizes the spatial resolution, which depends strongly on the local incidence angle (given by
the local slope angle η) and the topography because of the slant imaging geometry of SAR sensors (See Appendix). From the final LRW images, we estimated an effective resolution of about 15 x 25 m.

For avalanche detection we mapped areas which showed an increased radar backscatter signal in the difference of the pre- and post-avalanche event LRW-image. To remove bias by changing snow
properties (snow wetness), a 1 km high-pass filter was applied to the difference image. Additionally, to suppress noise but to preserve spatially structured details, a nonlocal mean filter (Buades et al., 2005; Condat, 2010) was applied.

## 3   Methods

To compare the different mapping methods, we proceeded in four steps (Figure 3) which are detailed in
the sections below:
1. Avalanches were visually detected or mapped based on the satellite data (Sect. 3.1), furthermore, we extracted mapped avalanches from an existing database, the Davos Avalanche Mapping project (DAvalMap; Sect. 3.3).
2. The ground truth data set was compiled from different sources (Sect. 3.2).





3. Validation points were created for all avalanches visible on ground truth photographs and, in addition, for all locations where at least one of the visual mapping methods showed an avalanche (Sect. 3.4). Properties (listed in the Appendix) were assigned to validation points i.a. describing which method detected an avalanche at the corresponding location.

4. Validation points located in areas not covered by the ground truth or outside the validation period were removed (Sect. 3.5).

5. Statistical measures were calculated (Sect. 3.5).

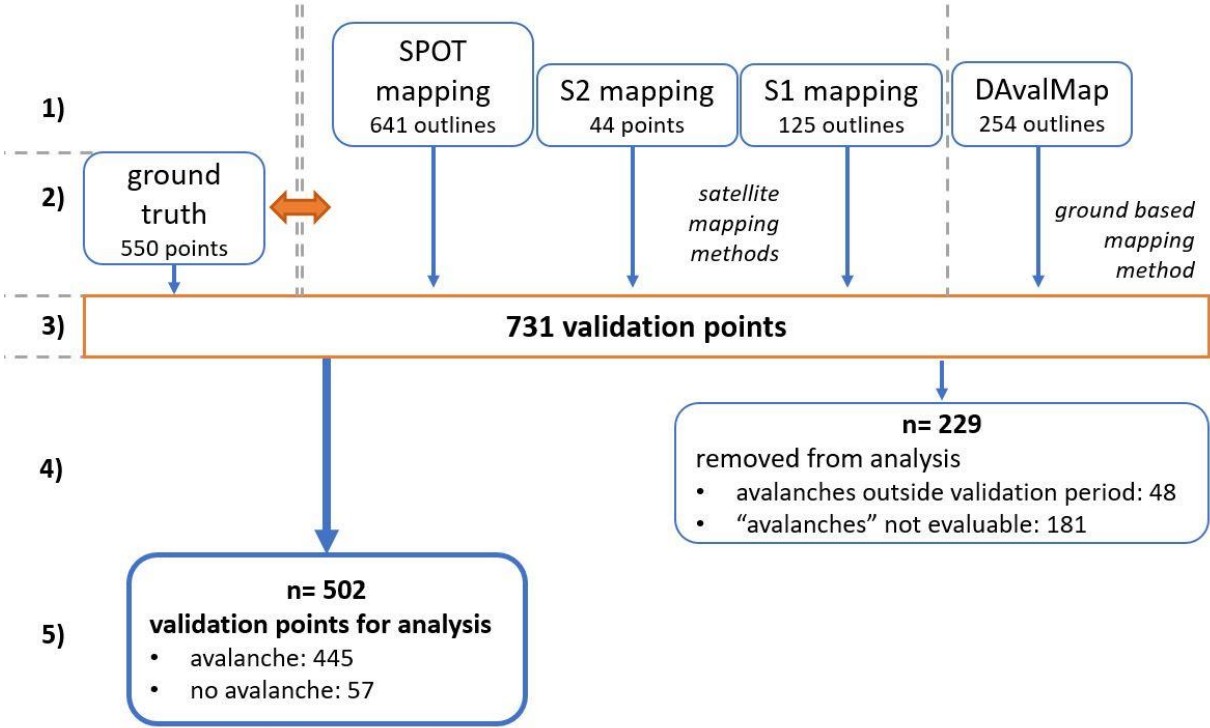

**Figure 3: Steps taken from the satellite mappings to the validation results. (1) Avalanches were mapped from satellite imagery and extracted from the DAvalMap database. (2) Ground truth data was compiled. (3) Validation points were created. (4) Points representing avalanches outside the validation period or outside ground truth were removed. The remaining validation points were used for analysis. The orange arrow symbolizes the link between ground truth and visually detected avalanches for validation.**

## 3.1 Visual detection of avalanches based on satellite data

For each of the three satellite image sources (Sect. 2.2), a different avalanche expert visually inspected the satellite images to detect and map features representing avalanches. We ascertained that the person mapping avalanches was familiar with the respective data-source as we experienced that a trained person achieved better results than someone without the specific training would. Furthermore, with a different person mapping the avalanches for each data-source, we prevented information leaking about

the presence of avalanches from one mapping method to another.



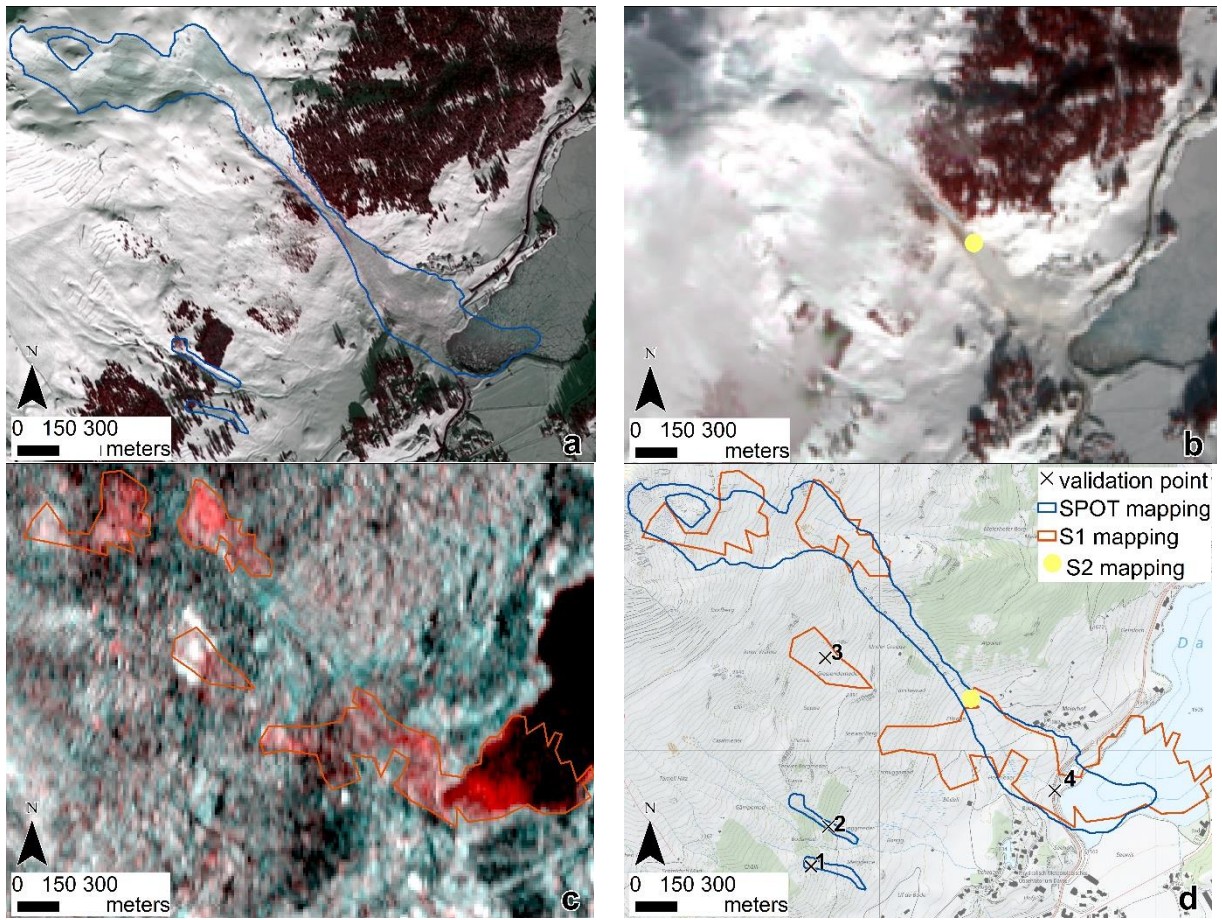

**Figure 4: Example of the mapping in 2019 and the base data used by the different methods for visual interpretation. (a): SPOT mapping (SPOT6 ©Airbus DS2019), (b): S2 mapping, (c): S1 mapping (Sentinel-1 and Sentinel-2 © Copernicus data (2019)). (d): Overlay of the three mapping results on the swissmap with the corresponding location of validation points. For validation point 1-3 a single mapped avalanche corresponds to one validation point, whereas for the validation point 4 multiple S1 polygons correspond to a single validation point (one-to-many join) generated from ground truth (Pixmap ©2019 swisstopo (5 704 000 000), reproduced by permission of swisstopo (JA100118)).**

**SPOT (SPOT mapping)**

We took advantage of the false-color band combination in near infrared (green, red, and near-infrared (NIR) band), where the reflectance of snow is lower (Warren, 1982). The mapping followed the methodology described in Bühler et al. (2019): avalanches were identified and digitized as polygons from optical images (Figure 4a). To improve visibility, image stretching, gamma optimization as well as modifications of contrast and brightness for separate outline digitization in the sun and shaded areas, were applied. Additional data like the Swiss Map Raster 25 (swisstopo, 2020b), the summer orthophoto mosaic SWISSIMAGE 25 cm (swisstopo, 2020a), and the layer "Slope angle over 30 degrees" calculated from the swissALTI3D (swisstopo, 2018), were used for interpretation. The mapping was performed as part of two verification campaigns following the avalanche-active periods in 2018 and 2019 (Bründl et al., 2019; Zweifel et al., 2019), conducted for a much larger area than our study area. Of all mapped avalanches 641 are located in our study area (2018: 523, 2019: 118).





**Sentinel-2 (S2 mapping)**

S2 mapping relied on false-color composite (green, red, and NIR) images (Figure 4b). For identification of avalanches, the post-event image was searched for identifiable avalanche features. Additionally, the pre-event image was consulted to identify changes (i.e. in forest) that might be connected to avalanches. As supplementary information, the SWISSIMAGE 25 cm (swisstopo, 2020a) was used. Avalanches were marked as points because the outline could not be meaningfully identified at the spatial resolution of S2. In total 44 points identifying avalanches were created (2018: 34, 2019: 10).

**Sentinel-1 (S1 mapping)**

For S1 avalanche polygons were mapped using the backscatter difference images (Sect. 2.2, Figure 4c). In uncertain cases (e.g. to remove bright pixels due to changing human objects), the radiometrically terrain-corrected backscatter images were considered for reference. In total 125 avalanche polygons were created (2018: 46, 2019: 79).

## 3.2 Ground truth

As ground truth, we relied on over 900 photographs taken before and after the two avalanche periods (Figure 2). Photographs were taken from the valley floor or from locations within the three ski areas by the interns of the avalanche warning service. Additionally, helicopters were used to document the exceptional avalanche activity. To avoid a bias from ground truth, we analyzed the ground truth not before finalizing the satellite mappings and the Davos avalanche mapping (Sect. 3.3). With plain photographs as ground truth, we could validate the existence of avalanches, albeit not the accuracy of outlines as outlines cannot be extracted from photographs.

Due to limited terrain visibility, our ground truth showed gaps for both validation periods where no validation was possible (Figure 5). Still, the available data allowed for validation of the majority of avalanches for each period as ground truth was available for 84 % of the perimeter in 2018, and for 74 % in 2019.





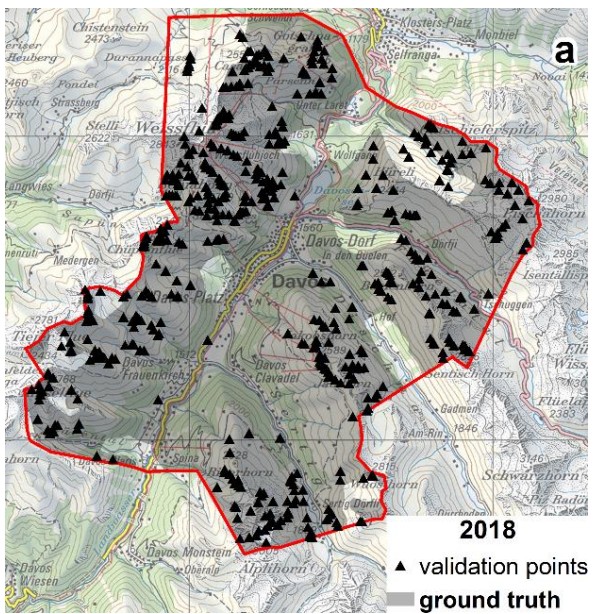
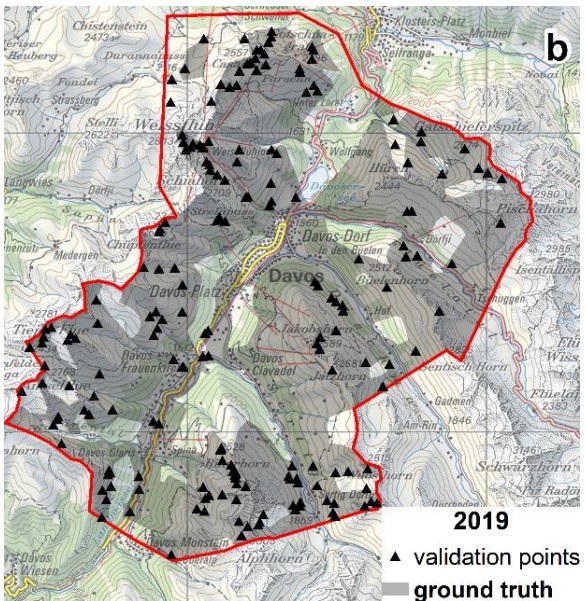

**Figure 5: Coverage of the study area with ground truth for (a) 2018 and (b) 2019 avalanche periods. The black triangles represent validation points. The points located outside ground truth could not be validated (Pixmap © 2019 swisstopo (5 704 000 000), reproduced by permission of swisstopo (JA100118)).**

**Avalanche size:** To relate the mapping results to avalanche size, we classified the avalanches at the
validation points. Two raters assigned avalanche size independent from each other using the ground
truth photographs. Avalanches were given one of five ordinal size classes (size 1 – small, size 2 –
medium, size 3 – large, size 4 – very large, size 5 – extremely large) according to the classification
defined by the European Avalanche Warning Services (EAWS, 2020;
https://www.avalanches.org/standards/avalanche-size/) or "unknown" if the size could not be
determined. The sizes assigned by the two raters corresponded well (Cohen's Kappa (Cohen, 1968) =
0.84, considered an *almost perfect agreement* (Landis und Koch, 1977)). For 56 cases, when avalanche
size differed (2018: 37, 2019: 19), the two raters discussed the size classification to assign a unique size.
For 79 % of avalanches one of the size classes could be assigned, the remaining 21 % of the avalanches
were classified as size "unknown".

## 3.3 Davos Avalanche mapping project (DAvalMap) – a ground truth alternative

Since the winter of 1949/50, avalanches occurring in the region of Davos are mapped. To obtain a high-
quality avalanche inventory, the national avalanche warning service, located at the SLF in in Davos,
cooperates with the rescue services of the ski areas and the council's avalanche warning service to
document avalanches. The area of the DAvalMap covers about 180 km$^2$ and corresponds to the study
area described in Sect. 2.1. Great efforts are made to obtain a complete-as-possible avalanche inventory.
However, missed avalanches and uncertain release dates may occur particularly during prolonged
storms with limited visibility or due to limited view of the more remote parts of the region.
Avalanches are recorded in the DAvalMap if the minimum extension is 50 m in one direction (width or
length) for slab or glide-snow avalanches, and a length of 100 m for loose-snow avalanches. Generally,



avalanches are documented by photographs taken in the field, and, at a later stage, their approximate outlines are mapped by the avalanche warning intern manually.

The Davos Avalanche Mapping (DAvalMap) data set is especially meaningful as it provides one of the rare data sets where avalanches have been mapped as comprehensively as possible for decades. The
DAvalMap data set has been used in several studies, e.g. for validation of the avalanche forecast, as input to model wet-snow avalanche occurrence and run-out distance, or to derive terrain characteristics describing potential release areas (e.g. Schweizer et al., 2003; Wever et al., 2018; Bühler et al., 2018; Harvey et al., 2018; Schweizer at al., 2020).

The properties of this data set make it a potential candidate to validate avalanches detected e.g. in
remote sensing time series. However, currently information about the quality and particularly the completeness of this dataset is missing, therefore we include it in the analysis and compare the DAvalMap with the ground truth dataset.

## 3.4 Validation points

To investigate the completeness of the avalanche mappings, we created validation points following the
steps specified below:
1. Validation points were created at locations where avalanches were detected on the ground truth photographs, as well as
2. for each avalanche polygon or point mapped in at least one of the mapping methods where the ground truth showed either no avalanche or where a human interpreter could not identify an
avalanche on ground truth sufficiently certain, additionally
3. to each validation point we assigned attributes describing which method detected an avalanche at the specific location (see also Appendix).

We placed no validation points in locations where no avalanche was detected, even though the
detection of non-events would have been correct. Validation points were either placed inside the area of the avalanche visible on the ground truth or, in case the ground truth showed no avalanche, or no ground truth was available, somewhere within the avalanche polygon of the corresponding mapping method. For matching locations, avalanches detected in the mapping methods had to be assigned to ground truth validation points. In most cases, a single avalanche - outline (SPOT, S1) or
point (S2) - was assigned to one validation point (Figure 6, a). However, as sometimes one avalanche was mapped with a single polygon by one method but split up into several polygons (or points) by another method, we allowed for one-to-many and many-to-one joins (Figure 6, b and c). A one-to-many join meaning one validation point being linked to multiple avalanche polygons, whereas many-to-one joins link one avalanche polygon to several validation points.




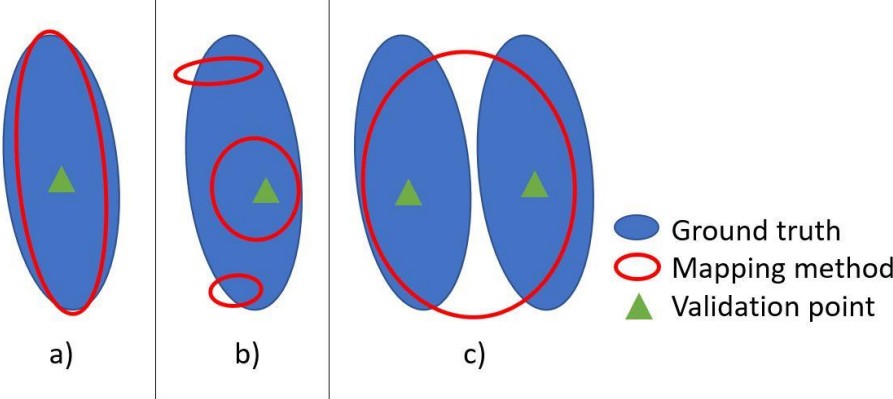

**Figure 6: Illustration of a one-to-one join (a), a one-to-many join (b) and a many-to-one join (c) used to assign the avalanches mapped by the different methods to the validation points.**

In total, we created 731 single validation points (2018: 534, 2019: 197). According to the ground truth,
we classified each point into one of three categories (see also Appendix): avalanche (1 - true), no
avalanche (0 – false) or validation not possible (2 - unknown). Of these, the 181 points classified as
unknown were omitted from further analysis (2018: 131, 2019: 50).
Orbit revisit times restricted the image acquisition times which differed by a few days as shown in
Figure 2. Therefore, it is possible that avalanches were mapped which had had occurred before or after
the validation period given by field photographs taken before and after the event. To remove them, 48
additional points were excluded (2018: 48; 2019: 2). This allowed us to validate in total 502 (73.5 %) of
all validation points (2018: 73.2 %, 2019: 74.4 %).

## 3.5 Statistical measures

To assess the detection performance of each mapping method, we calculated two statistical measures,
which are based on standard 2 x 2 contingency tables (Table 3):

**Table 3: Contingency table with two outcomes (avalanche = 1, no avalanche = 0; adapted from Trevethan, 2017).**

|  |  | mapping method | |
|---|---|---|---|
|  |  | 0 | 1 |
| ground truth | 0 | true negatives | false positives |
|  | 1 | false negatives | true positives |

To determine how many of the avalanches identified in the ground-truth were correctly detected by a
human on the satellite data (research question 1), we calculated the probability of detection (POD;
adapted from Trevethan, 2017):





$$POD = \frac{true\ positive\ avalanches}{true\ positive\ avalanches + false\ negative\ avalanches}$$

To determine how often was there really an avalanche when a human visually detected an avalanche on satellite data (research question 2), we calculated the positive predictive value (PPV; adapted from Trevethan, 2017):

$$PPV = \frac{true\ positive\ avalanches}{true\ positive\ avalanches + false\ positive\ avalanches}$$

### 3.6 Location-specific detection

**Avalanche illumination conditions in optical imagery**: Cast shadow on slopes has been observed to make avalanches difficult to detect in optical satellite imagery (Bühler et al., 2019; Leinss et al., 2020). Therefore, the SPOT avalanches were visually checked if the avalanche visible on ground truth was in fully illuminated, partly illuminated, or fully shaded parts of the SPOT imagery.

**Partial detection of avalanches by radar**: Among others, Leinss et al. (2020) and Abermann et al. (2019), pointed out that radar is more likely to detect the deposit area of avalanches whereas the release area and the avalanche track could often be missed. To assess this bias, we used the large number of avalanche polygons derived from Sentinel-1 in combination with the ground truth photographs to estimate which part of an avalanche is covered by the S1 avalanche polygon. For that we considered the upper third of the ground truth avalanche as release area, the middle part as avalanche track, and the lower third as the deposit area. For each covered part we added this information to properties of the corresponding validation point. Then we calculated the POD for the sub-set of S1 avalanches which contained only one of the three properties deposit, track, release area.

## 4   Results

We performed the following analyses:
  (1) Detection rate per size for each mapping method
  (2) POD and PPV of avalanches > size 2 for each mapping method
  (3) POD dependence on illumination for the SPOT mapping
  (4) Effects of partial avalanche detection in S1 mapping on POD and PPV
  (5) Implications of validation with other data as ground truth

According to the ground truth, 445 avalanches occurred in the two validation periods (2018: 318, 2019: 127). The resulting size distributions are shown in Figure 7. Except for size 1 avalanches, which we believe are under-represented in the ground truth, the observed size distributions agree with magnitude-




frequency distributions observed in other avalanche size distributions (i.e. Faillettaz et al., 2004; Schweizer at al., 2020).

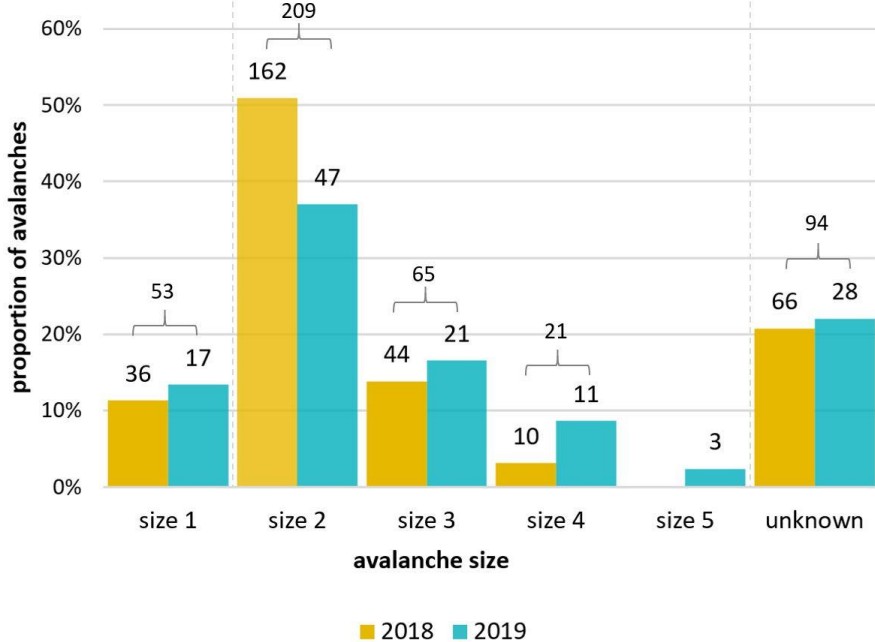

**Figure 7: Number and proportion of avalanches per size for 2018 (left bar) and 2019 (right bar). The proportions are indicated relative to the number of avalanches per year (2018: 318, 2019: 127). For the avalanches of unknown size, the raters could not reliably determine the size (Sect. 3.2).**

## 4.1 Avalanche detection rate per avalanche size (satellite methods)

Only the SPOT mapping approach detected all size 4 and size 5 avalanches (Figure 8a). The capabilities of the S1 mapping to detect the largest avalanches followed closely with 90 % of size 4 avalanches and all size 5 avalanches detected in 2019 (Figure 8b). By contrast, the S2 mapping only identified 29 % of size 4 avalanches and none of the size 5 avalanches in 2019 (Figure 8c). As Figure 8a-d illustrates all satellite methods show declining ability to map avalanches with decreasing size. This decline is more pronounced for the S1 than for the SPOT mapping. The S2 mapping identified very few avalanches altogether, especially for 2019.



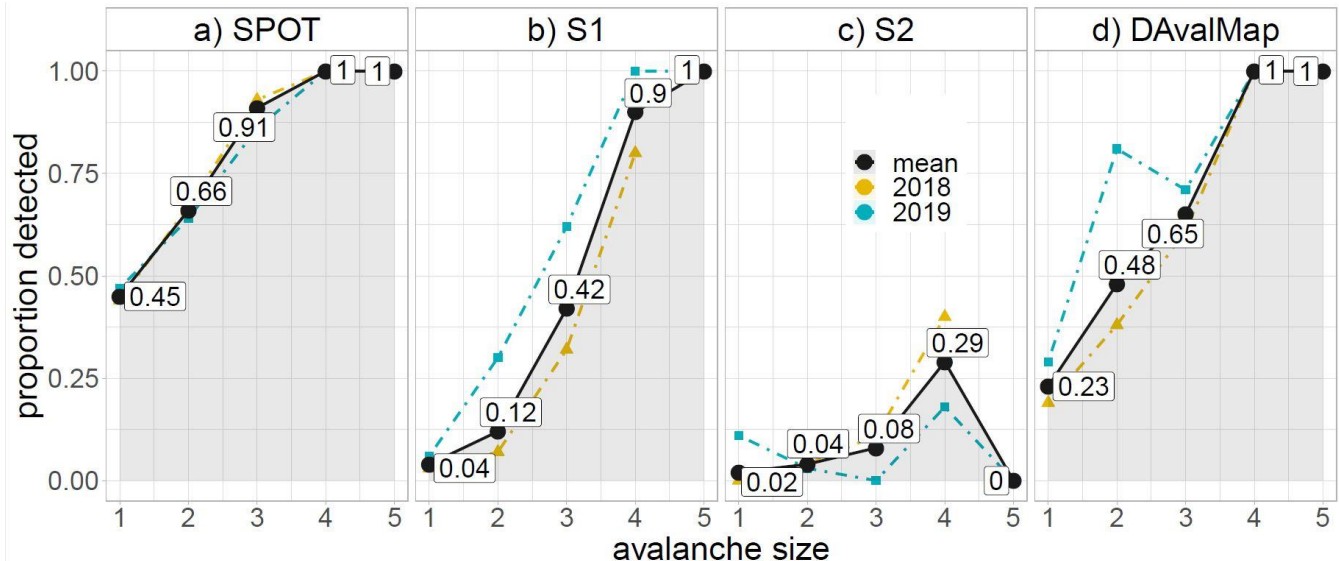

**Figure 8a-d: Detection rate by size for each of the avalanche mapping methods tested. The black dots and line represent the mean proportion of avalanches per size identified by the different mapping methods, additionally the proportions are shaded grey. In addition, the values for 2018 (yellow triangles) and 2019 (turquoise squares) are shown for each mapping method. For the number of avalanches in each size class and subset, refer to Figure 6.**

## 4.2 Detection statistics of the satellite mapping methods (POD and PPV, size >2)

Size 1 (small) avalanches are unlikely to cause damage or bury a person. Furthermore, they were probably also missed more frequently in the ground truth data. Therefore, in the following, we exclude size 1 avalanches and avalanches of unknown size and limit the analysis to the 298 avalanches confirmed by ground truth and classified as size 2 to size 5.

Avalanches of size 2 to 5 had occurred at 298 of the remaining 355 validation points (84 %), indicating that in 57 locations at least one of the methods falsely detected an avalanche. Considerable variations in the performance of the three satellite mapping approaches are noted:

- The probability of detecting an avalanche (POD), given its presence in the ground truth, varied greatly between methods (Table 4). Avalanches were most reliably detected by the SPOT mapping approach with 221 out of 298 detected avalanches (POD = 0.74), while the S1 mapping missed almost three quarters of the size 2 to 5 avalanches (POD = 0.27). Performance was extremely poor for S2 (POD = 0.06), highlighting that visual avalanche detection is nearly impossible in S2 data.
- The positive predictive value (PPV), the proportion of true positive avalanches to all avalanches mapped by a specific method, was greater than 0.8 for all methods (Table 4). Again, performance was best for SPOT (PPV = 0.88), and lowest for S2 with a PPV of 0.81, indicating that between one in five (S2) to one in nine (SPOT) mapped avalanches were false alarms.
- Comparing the performance between the two validation periods showed that the SPOT mapping is the most reliable one of the satellite-based methods with both performance metrics being similar in 2018 and 2019. The S1 mapping, in contrast, shows bigger differences between the





two validation periods, with the POD being clearly lower in 2018 (POD = 0.17) compared to
2019 (POD = 0.52), at least partly due to the larger occurrence of size 2 and 3 avalanches in
2018 (Figure 7 and Figure 8).

Table 4: POD and PPV of the different methods for the mapping from 2018, 2019 and together

|  |  | SPOT | S1 | S2 | DAvalMap |
|---|---|---|---|---|---|
| POD | 2018 | 0.74 | 0.17 | 0.07 | 0.46 |
|  | 2019 | 0.76 | 0.52 | 0.04 | 0.82 |
|  | all | 0.74 | 0.27 | 0.06 | 0.56 |
| PPV | 2018 | 0.87 | 0.90 | 0.88 | 0.90 |
|  | 2019 | 0.90 | 0.84 | 0.60 | 0.99 |
|  | all | 0.88 | 0.87 | 0.81 | 0.93 |

As illustrated in Figure 6 (Sec. 3.4), mapped avalanche outlines did not always correspond to one
validation point from ground truth, hence one-to-many and many-to-one joins were allowed (Figure 6).
Considering the SPOT and S1 methods only, the proportion of one-to-one joins was lower for S1 (76
%) compared to SPOT (88 %, Table 5). One-to-many joins, i.e. multiple detected avalanche patches
corresponding to one avalanche in the ground truth, were comparably frequent for S1 (14 %) and rare
for SPOT (3 %).

Table 5: Percentage of validation points where the specified joins were applied for the SPOT and S1 mapping. For each method only
the avalanches mapped and validated were considered.

|  | one-to-one | one-to-many | many-to-one |
|---|---|---|---|
| SPOT | 88 % | 3 % | 9 % |
| S1 | 76 % | 14 % | 10 % |

However, allowing one-to-many and many-to-one joins impacts results in two ways: firstly, in terms of
the correspondence between the number of features detected and the number of avalanches they
represent, and secondly it influences the calculated performance metrics (POD and PPV). For instance,
a method for which a high number of one-to-many joins was made (here S1), overestimates the total
number of avalanches, while it increases both POD and PPV. In contrast, a method characterized by a
high number of many-to-one joins tends to underestimate the number of avalanches assuming a one-to-
one translation between detected features and avalanches. Furthermore, many-to-one joins will decrease
both POD and PPV.




### 4.2.1 Effect of cast shadow on mapping from optical SPOT data

The detection rate using SPOT images depends strongly on whether the avalanche is located on a well-illuminated slope or in the cast shadow of surrounding mountains. The 221 avalanches correctly detected with SPOT mapping can be split into the following three categories:

- – In fully illuminated slopes, 127 of 147 avalanches were detected (POD = 0.86).
- – In partly illuminated slopes, 88 of 112 avalanches were detected (POD = 0.79)
- – In shaded slopes, 6 of 33 avalanches were detected (POD = 0.15)

Indicating a low detection rate for avalanches located fully in the cast shadow.

### 4.2.2 Partial avalanche detection in the S1 mapping

10 Avalanche polygons mapped in S1 data show, in comparison to SPOT data (Figure 4c vs. 4a), often multiple patches. These patches correspond to a single avalanche because parts of the avalanches are not visible in S1 imagery. The existence of multiple patches causes a discrepancy between the number of S1 avalanche polygons and the number of avalanches from ground truth and leads to the considerable number of 14 % of one-to-many joins (Table 5) whereas this number is relatively low (3 %) for SPOT.
15 However, we observed that the detectability of different avalanche patches depends on their relative location, i.e. to which part of the avalanche the patches belong to. According to the analysis described in section 3.6 we found that the total POD of 0.27 (Table 4) is reduced to a POD of 0.22 when only the deposit area is considered as done before by several authors (Abermann et al., 2019; Eckerstorfer et al., 2017; Lato et al., 2012). On one hand, this corresponds to the major part (75 %) of avalanches detected
20 by radar (Figure 9). On the other hand, however, 52 % of the radar mapped avalanches also mapped parts of the avalanche track and 35 % mapped even the release area. This, in turn, confirms the supposed bias toward the deposit area but highlights the importance to also map the release area and the avalanche path to obtain a better POD.

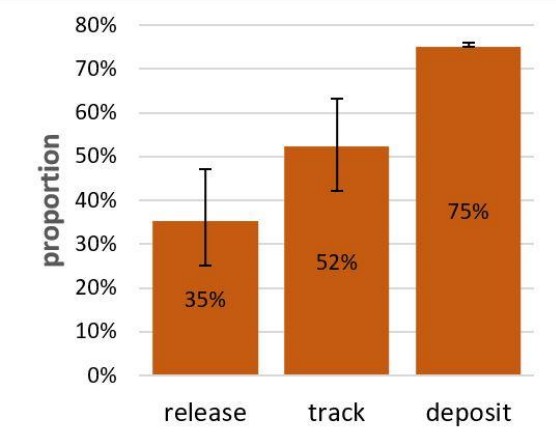





**Figure 9: Percentage of radar-detected avalanches where a feature was detected in the deposit, track, or release area of avalanches confirmed by ground truth photographs. The error bars indicate the respective proportions for the two validation periods.**

## 4.3 Validation with additional ground truth (DAvalMap)

In order to test alternatives that could be used as validation datasets in the future, we analyzed the
DAvalMap as a high POD in combination with a high PPV would indicate suitability. Analysis showed a high PPV of 0.93 for the DAvalMap, indicating a high reliability of mapped features in fact corresponding to avalanches (Table 4). However, POD was considerably lower with only about half of the size 2 to 5 avalanches being detected (POD = 0.56). As with the satellite mapping methods, the detection rate decreased strongly with decreasing avalanche size (Figure 8d). Furthermore, considerable
variation in POD between the two years was noted (2018: 0.46, 2019: 0.82). Performance metrics are generally more satisfactory for 2019, indicating a dependence on the person mapping.
In the following, we show the influence of relying on less complete data sets, like the DAvalMap, for validation on performance metrics of the satellite mapping methods (considering only the avalanches examined before in the analyses):
Recalculating POD and PPV relying on the DAvalMap as ground truth for SPOT inevitably affected PPV strongly; PPV decreased from 0.88 to 0.59. The comparably large number of SPOT true positive avalanches, considered false alarms according to DAvalMap, explain this. In contrast, the influence on POD is comparably small (0.74 to 0.78); as SPOT also detected many of the avalanches detected in the DAvalMap.
If we use the SPOT mapping as ground truth for the S1 mapping the POD decreases slightly from 0.27 to 0.24 with the PPV dropping from 0.87 to 0.73. Doing it the other way around, using the S1 mapping as ground truth for the SPOT mapping, the POD remains almost the same (0.74 to 0.73). In contrast, the PPV decreases from 0.88 to 0.24, caused by the large number of false positives, avalanches that were found in the SPOT but missed by the S1 mapping.

## 25  5   Discussion

### 5.1 Comparison of mapping approaches

We explored three mapping methods: optical, high-resolution SPOT; optical, lower-resolution Sentinel-2 (S2), and radar-based Sentinel-1 (S1). Of these methods, the 1.5 m resolution optical SPOT mapping achieved the best results for POD (0.74) and PPV (0.88; Table 4). It can detect avalanches of all sizes
(Figure 8a). The ~10m resolution S1 mapping in contrast, performs well for the identification of larger avalanches (size 4 or 5) but the overall POD is significantly lower (0.27) than for the SPOT mapping mainly because the majority of size 2 and 3 avalanches, which represent the largest number of all avalanches, were missed. The PPV of S1 (0.87) is in a similar range than SPOT.
Another quality aspect, which highlights SPOT mapping potential, is the high percentage of one-to-one
joins (88 %), indicating that the number of features detected by SPOT shows a closer correspondence





with the actual number of avalanches compared to S1 (one-to-one joins: 76 %). The results of the S2 mapping are poor with only one in 17 avalanches detected (POD of 0.06). We can therefore not recommend S2 for avalanche detection. Summarizing, high values of POD and PPV, in combination with a high proportion of one-to-one joins make a mapping with SPOT recommendable. However, in

the two situations explored, conditions were optimal for SPOT: the day immediately after the period of interest was cloud-free and satellite images could be obtained. This dependence of optical sensors on cloud-free conditions is the biggest disadvantage of the SPOT method. Additionally, SPOT data is costly and only available upon request.

Although the Sentinel-1 mapping achieved a considerably lower POD than SPOT, S1 permits

observations independent of weather and light-conditions. Furthermore, S1 data is free of charge and is operationally available (Table 6). Among others, Eckerstorfer et al. (2017) have focused on the mapping of avalanche deposit areas from Sentinel-1 imagery. As we have shown in Sect. 4.2, the deposit area could be identified for about 75 % of all avalanches by the S1 mapping. The remaining 25 % of S1 polygons captured release area and/or track only. This investigation indicates, that even though deposits

are more likely to be detected, the S1 mapping in many cases identifies other avalanche parts as well. Unfortunately, mapping results from S1 showed multiple patches corresponding to a single avalanche which need to be joined to avoid an overestimation of the avalanche number and an underestimation of the avalanche size. In order to solve this problem, an algorithm, joining S1 polygons belonging to the same avalanche, relying on terrain, would be desirable. We believe the automated snow avalanche

release area delineation from Bühler at al. (2018) may be adapted for such purpose.

With a POD of 0.06, Sentinel-2 imagery seem unsuitable for the mapping of avalanches. Abermann et al. (2019) found 23 % of avalanches on both Sentinel-1 and Sentinel-2 images, whereas we, in contrast, found only 9 % of avalanches from the S1 mapping overlapping with the S2 mapping. This might be due to better visibility of wet-snow avalanches, especially the slush flows in Abermann et al. (2019). An

overview over the strengths and weaknesses of all investigated satellite mapping methods is given in Table 6.

**Table 6: Summary of the strengths and weaknesses of the methods examined.**

| method | strength | weakness |
|---|---|---|
| SPOT mapping | • daily revisit capability due to constellation of SPOT-6 and SPOT-7<br>• may cover a very large area upon request (i.e. the whole Swiss Alps in one day)<br>• spatial resolution of 1.5 m well suited for avalanche detection<br>• visual avalanche identification is like what the eye is used to<br>• NIR band makes especially wet-snow avalanches well visible, no | • strongly depending on cloud-free conditions<br>• data only available if ordered and rather expensive (~100'000 US$ for an area of 12'500 km$^2$)<br>• if satellite is passing far from Nadir, high acquisition angles cause distortions in steep terrain (Figure 10)<br>• resolution of 1.5 m restricts the detection of size 1 and 2 avalanches |





| | | |
|---|---|---|
| | radiometric saturation on snow (Figure 10) | |
| S2 mapping | • orbit-revisiting time with the same acquisition angle every 5 days → covers large regions in several overpasses but regularly captures the same area<br>• image acquisition with incidence angles very close to zero<br>• data is free of charge<br>• visual avalanche identification is generally like what the eye is used to, but the spatial resolution is mostly insufficient | • strongly depending on cloud-free conditions<br>• resolution of 10 m very much restricts the visibility, even the mapping detection of size 4 and 5 avalanches is improbable |
| S1 mapping | • orbit-revisiting time 6 days- more often when combining data from different orbits<br>• acquisition in all weather and light conditions<br>• data is free of charge<br>• if ascending and descending images are combined the "blind spots" in layover and radar shade are negligible<br>• Sensitive to surface roughness changes such as avalanche debris<br>• spatial resolution well suited for detection of larger avalanches (≥ size 4) | • preprocessing computationally more expensive<br>• no mapping of avalanches in radar shadow and layover<br>• detection of size 1 and 2 avalanches is very limited, size 2 avalanches (50 – 200 m long) have an extension of just 2-10 pixels<br>• Avalanches are often only partially visible leading to overestimation of avalanche number and underestimation of the avalanche size |

Snow conditions differed between the two years: in 2018, both dry-snow and wet-snow avalanches had released, while in 2019 avalanches were dry (Figure 10). For the SPOT mapping we found no difference in the POD between dry and wet snow. In contrast, for radar-based mapping, it is commonly
5 reasoned that wet-snow avalanches are easier to detect (e.g. Leinss et al., 2020) which was confirmed in (Eckerstorfer et al., 2019) using ground truth data. Nevertheless, we obtained apparently the opposite result in the S1 mapping (POD and PPV better for 2019 with dry snow conditions; Table 4). This can be partially explained by the relatively large number of size 2 to 3 avalanches in 2018, which are more likely to be missed. Nevertheless, Figure 8b shows that during dry-snow conditions in 2019, a larger
10 fraction of size 2 and 3 avalanches could be detected, compared to the mixed-snow conditions (dry/wet) in 2018. Pre- and post-event radar backscatter images show much stronger overall changes of the snow conditions from mixed (pre-event) to wet-snow conditions (post-event) in 2018, whereas in 2019 with stable dry-snow conditions- avalanches were the most prominent changes of the backscatter signal. This, in turn, agrees with Eckerstorfer et al. (2019) who also observed a high POD for dry-snow
15 conditions in both (pre/post) images.



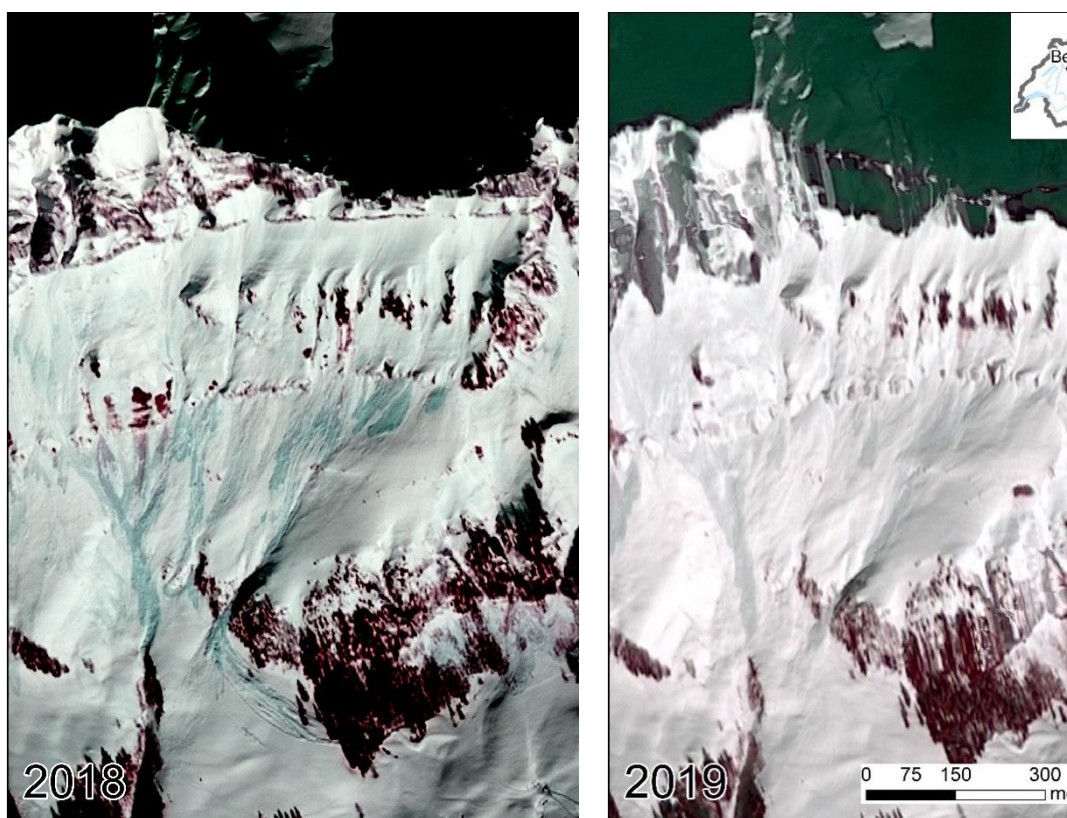

**Figure 10: In 2018, the temperatures and the snowfall line were high resulting in more wet-snow avalanches and deposits. Those are identifiable by the green shimmer in the NIR and more contrast to the surrounding snow in general (left side), which is not the case for the mostly dry-snow avalanches in 2019 (right side). Additionally, disparity in visibility in steep terrain due to different inclination angles is shown. The inclination angle for the tiles shown here lies at 11.3° in 2018 and at 27.6° in 2019. These distortions are among the worst we encountered comparing 2018 and 2019 SPOT data (SPOT6 data © Airbus DS 2018/2019).**

## 5.2 On the influence of the quality and definition of ground-truth on validation results

We showed the influence of using less complete avalanche observations as a ground truth alternative on the performance metrics (Sect. 4.3). Our findings are in line with theoretical investigations regarding the influence of errors in the reference class on POD and PPV (Brenner and Gefeller, 1997), but also with other studies outlining the importance of the definition of the ground truth on performance metrics (e.g. Techel et al., 2020 for snow instability tests).

As a specific example of a ground truth alternative, we relied on the DAvalMap data. However, the detection rate (POD = 0.56) clearly showed that this data set is far from providing a complete mapping. In fact, POD was lower for the DAvalMap compared to SPOT (POD = 0.74). In addition, differences in the quality of the mappings between the two years were large for the DAvalMap. The PPV has the highest value for the DAvalMap, in 2019 with 0.99 almost all avalanches mapped could be confirmed. These findings indicate that avalanches, which are stored in the DAvalMap data base, may be used for validation even though the mapping is partially inconsistent as already suspected by Schweizer et al.





(2003). However, to answer research questions for which comparably complete avalanche recordings are required, findings must be interpreted considering the uncertainty related to incomplete recordings. Both SPOT and Sentinel-1 data have been used previously to detect avalanches (e.g. Bühler et al., 2019; Eckerstorfer et al., 2019; Leinss et al., 2020). Each of these studies relied on a different ground truth:

Eckerstorfer et al. (2019) conducted a selective verification of 243 manually detected avalanches from Sentinel-1 imagery achieving a POD of 0.77. This is decisively better than the POD of 0.27 found for avalanches larger than size 1 in this study (Table 4). If we only consider avalanches larger than size 2, POD increases to 0.56 (while PPV drops to 0.79) – still considerably lower than the results presented by Eckerstorfer et al. (2019). We suspect that selective verification tends to overestimate POD, as in these

cases, verification data is usually available for well-visible prominent avalanches. This also showed in the slightly higher POD achieved for the SPOT mapping (Sec. 4.3), when relying on DAvalMap as ground truth, a ground truth which had a preference towards the detection of larger avalanches (Figure 8d).

Leinss et al. (2020) compared radar-detected avalanches (Sentinel-1) with optically detected avalanches

from SPOT (Hafner and Bühler, 2019). 68 % of the SPOT avalanches were detected by radar in their investigation. Inversely, 44 % of the radar-detected avalanches were detected by SPOT. In our study we linked mapped avalanches to validation points from ground truth. We found 89 % of the validation points representing avalanches larger than size 1, which were detected by the S1 mapping method, were also found in the SPOT mapping. In contrast, S1 detected only 55 % of the SPOT avalanches. Given the

validation with independent ground truth in this study, we believe our results to provide a more objective comparison of the two approaches.

Applying these findings to our study, we would argue that avalanches detected using SPOT images are a rather reliable ground-truth data source for slopes which are illuminated (or partly illuminated) and when sky conditions are clear. In contrast, if slopes are shaded or sky conditions do not permit good

visibility, SPOT images will be of little use for validation.

## 5.3 Strengths and limitations of our study

The study is limited to two avalanche periods. For these, we compiled a comprehensive ground-truth data set. However, beside our efforts to compile a spatially complete data set, we could not validate 48 detected features (2018: 38, 2019: 10) because of lacking ground truth, and 133 features (2018: 93,

2019: 40) because of low-quality ground-truth images. Furthermore, we expect that we missed some avalanches in the ground-truth images, particularly if these were of smaller size (see also Figure 7). Despite these limitations, we consider the ground-truth data to be complete enough to allow for a sound validation of detected avalanche features. Furthermore, the independently compiled ground truth data allowed for an objective comparison of the three satellite-based avalanche detection methods.

We explored just a small selection of the large number of potential satellite data sources, focusing on sensors and satellites previously used to detect and map avalanches (i.e. Eckerstorfer et al., 2019; Bühler at al., 2019; Leinss et al., 2020). Still, we consider the analyzed sensors and resolutions a representative selection of currently available satellite data sources. We relied on a human assessor to detect features representing avalanches visually. This approach depends heavily on the experience and

skills of the human performing the task (as has been shown for landslide-mapping, e.g. Hölbling et al.,





2015; Galli et al., 2008), and adds a certain degree of subjectivity to the analysis. Furthermore, manual detection of features is resource- and time-consuming.

To reduce the impact of limited visibility due to adverse weather and due to variations in operator performance, we suggest that future ground-truth data sets should be complemented with avalanche
occurrence data relying on automatic avalanche detection approaches, as for instance seismic or ground-based radar detection of avalanches (e.g. van Herwijnen and Schweizer, 2017; Mayer et al., 2020). Furthermore, recent advances in (semi-) automatically detecting avalanches are promising alternatives to complement avalanche occurrence data (Leinss et al., 2020; Eckerstorfer et al., 2019; Korzeniowska et al., 2017).

**6   Conclusions and outlook**

For the first time, we presented a spatially continuous, extensive validation of methods detecting avalanches from selected satellite imagery. We analyzed two avalanche periods for an area covering approximately 180 km$^2$ around Davos, Switzerland. We examined the potential, the advantages, and the disadvantages of the evaluated methods to provide decision guidance for those wanting to
comprehensively map avalanches in the future. We statistically confirmed several observations from Bühler et al. (2019) and Leinss et al. (2020): the SPOT mapping misses size 1 (small) and size 2 (medium) avalanches in several cases. S1 mapping misses most of size 1 and size 2 avalanches and over half of size 3 avalanches. We also confirmed that avalanches located completely in the cast shadow are much more likely missed, even on high resolution optical imagery (SPOT). For S1 we showed that
avalanche deposits are the avalanche part most likely detected, but the starting zone and the avalanche track are mappable in more cases than previously suspected.
The SPOT mapping holds great potential for comprehensive mapping of avalanches, at least for selected events for which costly and analysis-intensive SPOT data provides very valuable mapping results. The S1 mapping is quite reliable for larger avalanches (size 3 to 5) and allows for frequent and even
operational mapping for which automatic methods are currently developed (e.g. Eckerstorfer et al., 2019). Still it must be kept in mind that often the true size is underestimated, and that avalanches can appear partitioned into small patches which need to be joined by an advanced detection algorithm to estimate the true size. We found that Sentinel-2 data has a too low resolution to reliably map avalanches. Additionally, we explored the influence of ground truth on the validation results and
ascertained that incomplete, but otherwise reliable, ground truth datasets tend to overestimate POD and underestimate PPV.
We found that already existing satellite data provide great potential to approximate the avalanche activity and to get an overview of the spatial distribution of avalanches. However, for studies which require a precise and complete mapping of avalanche outlines further investigations are necessary. As
ground truth for such an examination unmanned aerial systems (UASs) were found to be a promising solution (Eckerstorfer et al., 2016; Bühler et al., 2017). To bypass time consuming manual mapping, an automation should be aimed at by developing reliable automated mapping algorithms or refining those that have already been created (Bühler et al., 2009; Lato et al., 2012; Korzeniowska et al. 2017). Prior to operational use of any approach, a comprehensive, not only a selective validation should be strived for.



For methods that have been comprehensively validated, the DAvalMap data base or a SPOT mapping might be used for selective follow-up validations.

**Data availability**

The datasets used in this study will be published in ENVIDAT (https://www.envidat.ch) with the final
publication of this study.

**Competing interests**

The authors declare that they have no conflict of interest.

**Author contributions**

EH performed the SPOT mappings, collected the ground truth, analyzed the datasets, assigned ground
truth size, coordinated the study, and wrote the paper draft. FT mapped from Sentinel-2 for 2018,
assigned ground truth size, delivered the necessary input from the SLF avalanche warning team,
critically reviewing the results, and heavily contributing to the paper draft. SL performed the Sentinel-1
data preparation and mappings, wrote the description thereof and reviewed and complemented the
manuscript with YB and FT. YB, FT and EH initiated the study together.

**Acknowledgments**

We thank all the ground truth photographers at the SLF and everyone else who helped to cover the area
around Davos as best as possible. We thank the Swiss avalanche warning service for providing the
Davos avalanche mapping outlines as well as for valuable feedback and support. We thank Linda
Zaugg- Ettlin for conducting the Sentinel-2 mapping for 2019. We thank the BAFU and the cantons of
Valais, Grison and Glarus as well as Liechtenstein for partly financing the SPOT mapping. We thank
Mathias Zesiger and Francesco Wyss from swisstopo for the processing of SPOT data and for
complementing its description. We are grateful to ESA for providing Sentinel-1 and Sentinel-2 data.



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





## Appendix

Slant image geometry (Sentinel-1)

In the study area, the incidence angle for both orbits is about $\theta = 42°$ (+-0.5°) resulting in a theoretical
5  ground range resolution of 4.6 /sin (42 + $\eta$) x 22 m = 6.9 x 22 m for horizontal terrain ($\eta = 0°$). For
slopes facing away from the sensor ($\eta > 0$) at a grazing incidence angle (42 + $\eta$ close to 90°), the
resolution could theoretically be improved up to the full slant range resolution of 4.6 x 22 m, but the
actual resolution is slightly reduced due to image interpolation. Larger slope angles result in radar
shadow and are not observable. On the opposite valley side, where slopes are facing towards the sensor
10  ($\eta < 0$) the resolution is significantly reduced. Slopes steeper than the incidence angle ($\eta < -42°$)
collapse into radar layover where non-adjacent areas get projected into the same radar image pixels such
that all resolution is lost. As the simulated backscatter intensity shows a large dynamic range which is
proportional to the area illuminated by the radar, LRW "selects" always the best resolution or averages
the backscatter signal if the resolution of both orbits is the same.

**Properties and required attributes for each validation point.**

| ID SPOT mapping | ObjectID of the avalanche mapped from SPOT satellite imagery, 0 if no avalanche is mapped | | |
|---|---|---|---|
| ID S1 mapping | ObjectID of the avalanche mapped from Sentinel-1 satellite imagery, 0 if no avalanche is mapped and -1 if the whole avalanche is in radar shadow/ layover | | |
| ID S2 mapping | ObjectID of the avalanche mapped from Sentinel-2 satellite imagery, 0 if no avalanche is mapped and -1 if the avalanche is located in clouded areas | | |
| name ground truth photograph | full name of the photograph on which this area is depicted, might include several images | | |
| validation | based on the photographs the following classification is performed: | | |
| | FALSE | 0 | no avalanche occurred at this point |
| | TRUE | 1 | at this point an avalanche occurred |
| | UNKNOWN | 2 | it cannot be said with sufficient confidence whether the avalanche mapped really occurred or not (the photographs are too low in resolution or inexistent) |
| avalanche size | 1 | size 1 | Small avalanche |
| | 2 | size 2 | Medium avalanche |
| | 3 | size 3 | Large avalanche |
| | 4 | size 4 | Very large avalanche |
| | 5 | size 5 | Extremely large avalanche |
| | 10 | size unknown | - |
| Illumination SPOT | Shade YES | | The avalanche is located fully in the shade. |
| | Shade NO | | The avalanche is located fully in illuminated terrain. |





| | Partly shaded | The avalanche is located in partly illuminated and partly shaded terrain. |
|---|---|---|
| **S1 avalanche part** | Parts of the avalanche (release, track and/or deposit) that were captured by the S1 mapping | |
| **comment** | supplementary information is put here:<br>• the ID of the other avalanches if several mapped avalanches were joined to one validation point ("one to many join")<br>• information if the avalanches were snowed upon and hard to see in the photographs<br>• if there wasn't an avalanche mapped with any of the methods but ground truth indicated the existence of one | |

Description of POD and PPV:

POD is the probability of the identification of a characteristic feature in the presence of such a feature (Brenner and Gefeller, 1997). In our case, the POD is the probability of an avalanche being mapped by a method given its existence.

PPV is the probability of the existence of a feature given its detection (Brenner and Gefeller 1997). A high PPV is desirable as it implies that false positive outcomes are minimized (Trevethan, 2017). In our case, PPV is the probability of the existence of an avalanche according to the ground truth given a mapping. If the PPV is high, avalanches mapped are likely to be "real" avalanches and falsely detected avalanches are kept to a minimum.