# Peer review of "Mapping avalanches with satellites – evaluation of performance and completeness"

_The Cryosphere, 2020_

## Referee Comment (RC1) · Anonymous Referee #1 · 2 Dec 2020

General comments:

The authors present a comparative study of the results of avalanche debris detection over 2 periods (20 to 24 January 2018, 13 to 16 January 2019) of high avalanche activity using SPOT-6, Sentinel-2 and Sentinel-1 radar images over an area of 180 km2 around Davos. The authors also use manually mapped avalanche data sets derived from photographs taken from the ground and from helicopters. The studied topic is of high interest for the scientific community since mapping avalanche debris and monitoring the subsequent avalanche activity is a very important issue in mountain regions. The authors therefore present general statistics on the detection of avalanche debris for each of the satellite observations used. They show the significant potential of very high resolution SPOT measurements and Sentinel-1 measurements for monitoring avalanche activity and the limitations of using Sentinel-2 for this application.

Overall, the manuscript is well written, the methodology is well described and the objectives of the study are well defined. This paper therefore deserves to be published, but I suggest that the authors make some improvements to the paper according to the comments and questions I list below. Some points would require additional information for the benefit of the readers.

Specific comments:

- Adding a processing scheme of the SAR data processing from Ground Range Detected High Resolution images to local resolution weighting images would be very useful. - It is a very good option to use radiometrically flattened and terrain-geocoded SAR observations according to the methodology proposed by Small (2011). How the difference in observation time between the pixels of the ascending and descending image is handled ? and what about the differences in observation angle ? Would it be better to merge images in this way or to keep the ascending and descending orbits separated (but still corrected for the radiometric effects of topography) ? And then merge the binary detected avalanche debris pixels ? - For avalanche detection, areas that show an increase in the radar backscatter signal in the difference of the LRW image before and after the avalanche event are targeted. So what is the threshold you used? Did you looked at the signal variation in an observed avalanche corridor to validate the choice of threshold ? - Once the pixel detection is done, I do not really understand how you go from the detected pixel to a detected event ? This is an important step because it is more relevant in my opinion to look at detections in terms of events rather than pixels. - Given the test area, some other SAR images would be suitable from different ascending/descending orbits (A117, D129, ) in addition to A15 and D168. - The month of January 2018 was exceptional in terms of avalanche activity and avalanches that were recorded on January 24 (In SPOT) may in fact have occurred earlier in the month. How can this effect be taken into account? Have you filtered the events to retain only the most recent ones? - As you mentioned in the paper, there are

areas that are less well observed using SAR measurements. And this could induce a bias in the detection of debris depending on the orientation of the slope. Have you looked at the results of detections by main orientation? - Similarly, given the difference in observation time between ascending and descending orbits, one would also expect a noticeable difference in detection results between the two orbits, which also argues in favor of separating the morning and evening orbits. - Regarding contingency tables, while the notion of true positive is simple to elaborate, the notion of false negative is more questionable. Because outside the ground truth, it is difficult to say if a satellite detection is "false" (difference in observation time, rain/snowfall after event, . . .). - Is it possible to explain the detection difficulties with Sentinel-2? Is it a matter of information content or pre-processing or band selection? - Regarding SAR weaknesses, I think that more effort should be put on methodologies to better isolate avalanche debris signals in images (adaptive thresholding depending on the type of surface, or efficient image analysis methods to detect signal change). These data are rich in information but unfortunately complex to use in the absence of an open and ready-to-use database of pre-processed Sentinel-1 measurements for scientists. - Finally I would like to thank the authors in advance for sharing the data from this work as it was the case following the article by Bühler et al. 2019. I had the pleasure of using this validation data and it was decisive in developing our avalanche debris detection algorithms in France.

---

## Referee Comment (RC2) · Markus Eckerstorfer (Referee) · 2 Dec 2020

Dear authors, dear editor!

Thank you very much for the opportunity to review this manuscript, I enjoyed reading it. This is another important contribution to the still small but quickly evolving field of remote sensing of snow avalanches and the authors very well explain the applied importance of this scientific field. Due to the field being relatively small, meaning that more or less the same groups publish studies, we also seem to review each other's papers all the time. I raised that concern to the editor when I accepted this review and I think it is still valid. More so because from the review of referee #1, I can guess to which group he/she belongs to. While it is certainly of advantage for the improvement

of the manuscript that experts review it, I would very much welcome if avalanche or remote sensing experts would also have a go, possibly bringing a fresh mindset to the table. Having said that, here is my review:

I agree with the authors that it is timely that a detailed evaluation of avalanche detection using remote sensing is carried out and I also agree with the authors that they got the datasets to do so. I therefore recommend publishing this article, however, would like to suggest some potential improvements:

Major concerns: I have two major concerns, where 1) deals with the limitation to study only two single avalanche events in a small area, using only a handful of satellite data and 2) is the methods section and my understanding of what you are doing.

1) I understand that such a detailed evaluation is only possible with focusing on single avalanche events where complete ground truth data is available. I am therefore hoping that the authors elaborate more on the limitations of their study in terms evaluating detection performance in the different sensors using only 1-2 images. We know that the SAR signal from snow is highly dynamic, influencing avalanche detection. Illumination as you suggest in the manuscript plays an important role for avalanche detection in optical imagery.

2) I am not fully able to follow your method section, especially the schematic of Figure 3. I think Figure 3 as well as Figure 4 are very important for reproducibility as well as comprehension of your study. Here are my detailed questions about the method section and these figures: Ground truth: Does 550 ground truth points mean that 550 avalanches were used as ground truth? Where in the avalanches were these ground truth points located and could you show them in Figure 4. Or are these actually polygons as indicated in Figure 6? Validation points: 731 validation points were used according to Figure 3, whereof 550 stem from ground truth points and the rest stems from where? How did you choose the location of these validation points inside avalanches and does this location have an affect on your joining you show in Figure 6? Figure 3:

Does this Figure have a time scale on its 'x-axis? In other words, does it show the sequence of steps or not? What does the stapled line with the orange arrow mean? Figure 4: a) Would it be possible to show slopes over 30 degrees? B) Could you maybe show these changes prior – post event that you are talking about? C) Could you explain the colors in the backscatter difference image and would it be worth showing the single backscatter image that is sometimes used? How was discerned if multiple features where actually from a single avalanche? Could you show release, track, deposition area? Could you indicate the time of all these images. Figure 6: You only show cases where you have a detected avalanche outline. How did the joining work for Sentinel-2 where you created only a point instead of a polygon? How was the spatial joining done? Did you do that manually or automatically with some sort of spatial buffer? How close would a validation point and detection be to be joined? Did you only consider spatial overlaps? Again, how would the setting of a validation point affect the joining?

Minor comments: Table 5: could it be an idea to calculate POD and PPV for one to one and one to many joins? I got curious how that would play out especially for Sentinel-1.

4.2.1 Could you calculate how much of the area is affected by fully and partly illumination and shade areas and discuss how that would change throughout a winter (min, max). This plays back to my major concern about evaluating only single events.

---

## Author Comment (AC1) · 16 Dec 2020

Dear Markus Eckerstorfer,

thank you very much for your constructive and careful review of our paper. We totally agree that it's only three or four groups of independent scientists reviewing each other's papers about avalanche detection. We also agree with you that reviewers with another focus in remote sensing or avalanche experts would enrich the discussion and help the topic make progress with new ideas. However, in our point of view it is essential that the review is performed by someone who have the necessary technical skills but also a background in application to judge the full value of the new findings. Unfortunately, there are not so many specialists available today that fulfill these prerequisites. Please

find in the following our answers to the issues and questions you raised:

1) Limitations of this study:

We are aware that drawing conclusions about the overall accuracy of a method our "small" investigation has certain limitations that we attempted to carefully lay out in 5.3. In optical data, of course given a cloud-free image, illumination seems to be the most important factor for accuracy (4.2.1). Calculations have shown that 65% (61%) of the investigated perimeter were illuminated at the time of SPOT image acquisition in 2018 (2019). In order to do the effects of illumination conditions in optical data justice, we will add a small section to 4.2.1 and the discussion debating the implications of changing illumination conditions on the area affected by cast shadow over the course of the winter and its expected effects on the mapping accuracy.

As for radar data, the orbits were chosen because the far- range minimizes layover and improve avalanche visibility (see Leinss et al., 2020), we were not thinking about SAR signal change due to snow conditions as we selected images. But, we have pointed out in 5.1 that we observed that pre-and post-event radar backscatter images show much stronger overall changes of the snow conditions from mixed (pre-event) to wet-snow conditions (post-event) in 2018, whereas in 2019 with stable dry-snow conditions-avalanches were the most prominent changes of the backscatter signal. Additionally, we have pointed to the investigation of a series of SAR images exploring snow conditions in Eckerstorfer et al.(2019). We believe elaborating the effects of snow conditions on SAR imagery is beyond the scope of this paper, as the paper's objective is the comparison of different sensors rather than an in-depth study of radar-specific properties. As you pointed out, a detailed evaluation has been due, and we hope that more such investigations will follow. Therefor we have carefully described the applied data and methods and believe that with further comparison it will be possible to better assess the accuracy under varying conditions, in different regions and in diverse terrain.

2) Ground truth clarification:

As our ground truth was not comprehensive (Figure 5) we could not join avalanches identified in ground truth with those mapped in the different methods and automatically declare those without a match a false detection. We therefor had to examine our ground truth twice: first to identify avalanches and create validation points which we continued to match with the mapped avalanches. Second to check whether for the remaining unmatched avalanches (from our examined methods) we had ground truth and could proof a false detection or if the validity of that mapping would have to be declared unknown (which is where the remaining validation points come from).

Therefore, after identifying avalanches (i.e. validation points in ground truth), matching them with our avalanches from the satellite and ground based mapping methods, and backchecking for false detections, we had 550 points with valid information from our ground truth images. As mentioned, this includes confirmed avalanches as well as confirmed false detections. 48 of those 550 validation points where identified as representing avalanches outside our validation period, therefor only the remaining 502 went into analysis. The number of avalanches mapped in either method that could not be evaluated is 181, which together with 48 avalanches outside the validation period accounts for 229 avalanches that were not considered for analysis. We will attempt to make the above procedure even more clear in the final version of our paper.

Validation points:

The validation points were manually created in a location overflown by the avalanche based on the ground truth photo. We did not follow a pattern in which fraction (release, track, deposit) of the avalanche the validation point was placed. In Figure 4 the validation points are already shown in 4d. The validation points are actually points, but as we wanted to illustrate the relationship between the avalanches in the ground truth photographs with the mapped avalanches we chose to show joins with polygons in Figure 6. The location of our validation points did not have an effect on the results of joining

as we always went back to the corresponding ground truth photograph for clarification in case of ambiguity.

Figure 3:

The numbers to the left represent a sequence of steps, in that sense an x-axis. As noted in the legend, the orange arrow symbolizes the link between ground truth imagery and visually detected avalanches for validation.

Figure 4:

We will add aspect to Figure 4d in the final version of this paper, but not for Figure 4a-c as we are convinced it would considerably worsen the readability. Additionally, we will indicate the time of acquisition for the imagery in 4a-c in the legend of Figure 4. Whether or not multiple features belong to the same avalanche was decided relying on the ground truth photographs. We cannot show release, track and deposit as we do not know the exact dimensions of those avalanche parts (UAV data could provide that). If you are referring to 4.2.2 with this request, release, track and deposit were defined as upper, middle and lower third of the avalanche shown on ground truth imagery. As for the backscatter of SAR in 4c: red represents the post-event image, green/blue the pre-event image. The image is composed by LRW using the four Terrain-Corrected images from 2019 as listed in Table 2. We never used single-backscatter images. We did the detection based on the backscatter ratio (not shown) and looked at the shown LRW/TC corrected image in unsure cases. We will add the explanation of the colors shown in 4c to the legend in the final version.

Figure 6:

We tried to show spatial joins with respect to "reality" which is why we decided to picture outlines for ground truth avalanches. The spatial joining of validation points to avalanches mapped by either of the methods was done manually. As we could always go back to the original ground truth photographs depicting the avalanches on which

our validation points were based on, proximity of points was never a deciding factor for joins. The avalanche from either mapping method had to overlap with the avalanche visible in the ground truth in order to be joined to the corresponding validation point. Hence, joining Sentinel 2 to the validation points did not pose a problem. Summarizing, because we looked at the underlying ground truth photographs in unclear situations, the position of the validation points did not have an effect on the joining procedure.

Table 5:

Based on your request we calculated POD and PPV neglecting joins. For the computation we treated multiple mapped avalanche patches which were originally joined to one validation point as separate avalanches (one-to-many) and one avalanche patch as just one avalanche even though it was joined to two validation points because of avalanches on ground truth (many-to-one). In order to make the effects of either join better visible we have calculated them both separately and together. The results are depicted in Figure 1 at the end of this document.

It can be seen that treating several avalanche patches as several avalanches (using no one-to-many joins), overestimates the number of avalanches, leading to a higher POD and PPV. Compared to the numbers in Table 4, the increase in POD for S1 is more pronounced as the percentage of one-to-many joins is higher (Table 5). If we are neglecting many-to-one joins and treating one avalanche polygon as one avalanche (even though ground truth showed two or more corresponding avalanches) the POD decreases as well as PPV. If both one-to-many and many-to-one joins are neglected, for SPOT the POD and PPV are slightly lower than the results in Table 4, whereas the opposite is true for S1. This is due to one-to-many joins being more relevant for S1 and many-to-one joins for SPOT. We will add this explanation to the Appendix of the final paper and add a few sentences to the section referring to joins in 4.2.

4.2.1

We think there is a misunderstanding about avalanches in partly illuminated terrain:

the category means that part of the avalanche (at least one 5th) is located in shaded and the remains in illuminated terrain (at least one 5th). As mentioned in 1. we will add a section discussing the change in shaded and illuminated areas over the course of the winter and its implications for the results in the final version of our paper.

Eckerstorfer, M., Vickers, H., Malnes, E., and Grahn, J.: Near-Real Time Automatic Snow Avalanche Activity Monitoring System Using Sentinel-1 SAR Data in Norway, Remote Sensing, 11, 2863, doi:10.3390/rs11232863, 201

Leinss, S., Wicki, R., Holenstein, S., Baffelli, S., and Bühler, Y.: Snow Avalanche Detection and Mapping in single, multitemporal, and multiorbital Radar Images from TerraSAR-X and Sentinel-1, doi:10.5194/nhess-2019-373, 2020.

| | no one-to-many joins | | no many-to-one joins | | no joins at all | |
|---|---|---|---|---|---|---|
| | SPOT | S1 | SPOT | S1 | SPOT | S1 |
| **POD** | 0.75 | 0.31 | 0.70 | 0.25 | 0.72 | 0.29 |
| **PPV** | 0.89 | 0.89 | 0.86 | 0.86 | 0.87 | 0.88 |

**Fig. 1.** POD and PPV calculated neglecting one-to-many, many-to-one and both joins.

---

## Author Comment (AC2) · 16 Dec 2020

Dear reviewer #1,

thank you very much for your comments and suggestions for our paper. Below we provide specific answers to the processing and analysis of Sentinel-1 data. However, we like to emphasize that the paper's objective is the comparison of different sensors rather than an in-depth comparison of different radar-specific methods. For this reason, and because the paper has already a considerable length, we consider several aspects (and questions) to be beyond the scope of this paper. For a more detailed description of SAR data processing we refer to Leinss et al. (2020).

RC: Adding a processing scheme of the SAR data processing from Ground Range

[Figure]

Detected High Resolution (GRDH) images to local resolution weighting images would be very useful.

- Thank you for this valuable point. As input we used SLC not GRDH images. We will add this information to Table 2: Mode = (IW, SLC). Please find at the end of this comment the complete processing flow chart for processing of the radar images. We will add this flow chart to the appendix of the final paper. See also https://forum.step.esa.int/t/sar-simulation-terrain-correction-no-output-of-simulated-intensity/23513/12.

RC: It is a very good option to use radiometrically flattened and terrain-geocoded SAR observations according to the methodology proposed by Small (2011). How the difference in observation time between the pixels of the ascending and descending image is handled?

- We use terrain flattening according to Small (2011) and used the simulated SAR image as a weight for LRW according to Small (2012) as described in the paper. The difference between ascending and descending image is 1.5 days. Because the main avalanche activity (with level 4 and 5, see Fig. 2) happened not within these 1.5 days, it is unlikely that avalanches have occurred within these 1.5 days. Instead, most of the detected avalanches must have occurred between the pre- and post-acquisitions. Furthermore, as the weight used for LRW is linear to the illuminated area (which is proportional to the (linear) backscatter intensity) the image composition follows rather an almost binary weighting (especially for non-horizontal terrain) than an equally weighted average (which is only used for nearly horizontal terrain). This makes LRW a good method for image composition and the chance to miss avalanches by averaging is very low. We will add these points to the discussion of our paper.

RC: ...and what about the differences in observation angle?

- In Leinss et al. (2020) we list reasons why the relative brightness of avalanches is stronger for slopes facing away from the radar. As these slopes are weighted stronger

by LRW, LRW enhances the visibility of avalanches. We will add this information to line 25, page 7.

RC: Would it be better to merge images in this [LRW] way or to keep the ascending and descending or-bits separated (but still corrected for the radiometric effects of topography)? And then merge the binary detected avalanche debris pixels?

- It depends on the application: In our case, LRW can reduce radar speckle and can therefore enhance the apparent spatial resolution which makes detection of smaller avalanches possible, when acquisitions are selected for a specific event. For an operational application, where acquisitions are not selected for a specific avalanche event, keeping ascending and descending separate allows for a slightly better temporal resolution in the detection. By merging the binary maps, the temporal resolution is lost again, therefore, if ascending and descending is merged anyway, we consider LRW as the optimal solution. We will add this information to the discussion.

RC: For avalanche detection, areas that show an increase in the radar backscatter signal in the difference of the LRW-image before and after the avalanche event are targeted. So what is the threshold you used? Did you looked at the signal variation in an observed avalanche corridor to validate the choice of threshold?

- As shown in the attached processing-graph, avalanches were manually detected based on the apparent visual brightness and the shape and size of bright pixels. No pre-defined threshold was used as the mapping was manually done. We will clarify this in line 10, page 13.

RC: Once the pixel detection is done, I do not really understand how you go from the detected pixel to a detected event? This is an important step because it is more relevant in my opinion to look at detections in terms of events rather than pixels.

- As we are interested in a size-resolved comparison, avalanches with S1 were mapped as avalanche polygons based on the visible area of bright pixels. In some cases,

polygons in the same avalanche corridor were joined to one event. See line 25-30, page 12 for more information.

RC: The month of January 2018 was exceptional in terms of avalanche activity and avalanches that were recorded on January 24 (In SPOT) may in fact have occurred earlier in the month. How can this effect be taken into account? Have you filtered the events to retain only the most recent ones?

- We agree that differing release and acquisition dates pose a challenge for an investigation as ours. In order to take these effects into account we rely on ground truth from different days in January (Figure 2). Through that we filtered the events for the desired validation period by excluding 48 confirmed avalanches from analysis as described in 3.4. Hence this effect of older avalanches distorting the results could be eliminated for 2018 (and 2019).

RC: Given the test area, some other SAR images would be suitable from different ascending/descending orbits (A117, D129) in addition to [the used orbits] A15 and D168.

- We chose the orbits A15 and D168 (IW3) because they were acquired in far-range (larger incidence angle, compared to A117 and D129: IW1). This minimizes layover and should make avalanches better visible (see Leinss et al., 2020). Note: In a paper - to be submitted soon - we have quantified and confirmed the incidence-angle dependent avalanche brightness.

RC: As you mentioned in the paper, there are areas that are less well observed using SAR measurements. And this could induce bias in the detection of debris depending on the orientation of the slope. Have you looked at the results of detections by main orientation?

- Such an analysis would be beyond the scope of this paper which focuses on the comparison of different sensors rather than a specific SAR analysis. But we propose

that the avalanche polygons published by Bühler et al., 2019 could provide a good basis for such an analysis.

RC: Similarly, given the difference in observation time between ascending and descending orbits, one would also expect a noticeable difference in detection results between the two orbits, which also argues in favor of separating the morning and evening orbits.

- As argued above, LRW selects the best visible slope from each orbit. This way, the areas visible from either ascending or descending orbits are analyzed in one go. Also, as argued above, we focused our analysis on one event, not on specific morning or evening events.

RC: Regarding contingency tables, while the notion of true positive is simple to elaborate, the notion of false negative is more questionable. Because outside the ground truth, it is difficult to say if a satellite detection is "false" (difference in observation time, rain/snowfall after event, . . .).

- We have excluded all mapped avalanches located outside our ground truth from the analysis as mentioned in section 3.4. As soon as ground truth is available, the determination of true positives, false positives and false negatives does not pose a problem.

RC: Is it possible to explain the detection difficulties with Sentinel-2? Is it a matter of information content or pre-processing or band selection?

- For Sentinel-2 we believe the detection difficulties are caused mainly by the spatial resolution. The pre-processing and band selection are similar to SPOT which has shown to give good results.

RC: Regarding SAR weaknesses, I think that more effort should be put on methodologies to better isolate avalanche debris signals in images (adaptive thresholding depending on the type of surface, or efficient image analysis methods to detect signal change). These data are rich in information but unfortunately complex to use in the

absence of an open and ready-to-use database of pre-processed Sentinel-1 measurements for scientists.

- There is significant effort put into processing methods of Sentinel-1 imagery for Avalanche detection at various research institutions in France, Norway, Switzerland. A detailed analysis of LRW for preprocessing including a database of radar-detected avalanches is planned, but is beyond the scope of this paper. In our opinion the apparent absence of ready-to-use pre-processed Sentinel-1 is of advantage because it provides the space for avalanche-specific pre-processing of SAR data, starting from GRDH or SLC data. "Analysis-Ready-Data" from Sentinel-1 for Switzerland might be available soon on https://www.swissdatacube.org/index.php/2018/12/05/first-sentinel-1-analysis-ready-data-ingested/.

Bühler, Y., Hafner, E. D., Zweifel, B., Zesiger, M., and Heisig, H.: Where are the avalanches?: Rapid SPOT6 satellite data 25acquisition to map an extreme avalanche period over the Swiss Alps, The Cryosphere, 13, 3225–3238, doi:10.5194/tc-13-3225-2019, 2019. Leinss, S., Wicki, R., Holenstein, S., Baffelli, S., and Bühler, Y.: Snow Avalanche Detection and Mapping in single, multitemporal, and multiorbital Radar Images from TerraSAR-X and Sentinel-1, doi:10.5194/nhess-2019-373, 2020.
* * *
[Figure]

**Fig. 1.** Processing Workflow for the SAR imagery.